# Identification of a Biomarker Combination for Survival Stratification in pStage II/III Gastric Cancer after Curative Resection

**DOI:** 10.3390/cancers14184427

**Published:** 2022-09-12

**Authors:** Itaru Hashimoto, Yayoi Kimura, Naohide Oue, Yukihiko Hiroshima, Toru Aoyama, Yasushi Rino, Tomoyuki Yokose, Wataru Yasui, Yohei Miyagi, Takashi Oshima

**Affiliations:** 1Department of Gastrointestinal Surgery, Kanagawa Cancer Center, 2-3-2 Nakao, Asahi-ku, Yokohama 241-8515, Japan; 2Department of Surgery, Yokohama City University, 3-9 Fukuura, Kanazawa-ku, Yokohama 236-0004, Japan; 3Advanced Medical Research Center, Yokohama City University, 3-9 Fukuura, Kanazawa-ku, Yokohama 236-0004, Japan; 4Department of Molecular Pathology, Graduate School of Biomedical and Health Science, Hiroshima University, Hiroshima 734-8551, Japan; 5Kanagawa Cancer Center Research Institute, 2-3-2 Nakao, Asahi-ku, Yokohama 241-8515, Japan; 6Department of Pathology, Kanagawa Cancer Center, 2-3-2 Nakao, Asahi-ku, Yokohama 241-8515, Japan

**Keywords:** platelet-derived growth factor receptor-beta (PDGFRB), inhibin subunit beta A (INHBA), matrix metallopeptidase-11 (MMP11), galectin-10, survival stratification marker, gastric cancer

## Abstract

**Simple Summary:**

Gastric cancer (GC) is the fifth most common cancer worldwide and the fourth most common cause of cancer-related deaths, with a high frequency of recurrence and metastasis, and a poor prognosis. This study presents a novel combination of four proteins (PDGFRB, INHBA, MMP11, and galectin-10) in GC tissues that have been identified as useful survival stratification markers in patients with pStage II/III GC after curative resection by quantitative polymerase chain reaction (qPCR), proteomic analysis, and immunohistochemistry (IHC).

**Abstract:**

Background: We sought to identify an optimal combination of survival risk stratification markers in patients with pathological (p) stage II/III gastric cancer (GC) after curative resection. Methods: We measured the expression levels of 127 genes in pStage II/III GC tissues of two patient cohorts by quantitative polymerase chain reaction (qPCR) and the expression of 1756 proteins between two prognosis (good and poor) groups by proteomic analysis to identify candidate survival stratification markers. Further, immunohistochemistry (IHC) using tumor microarrays (TMAs) in another cohort of patients was performed to identify an optimal biomarker combination for survival stratification in GC patients. Results: secreted protein acidic and rich in cysteine (SPARC), erb-b2 receptor tyrosine kinase 2 (ERBB2), inhibin subunit beta A (INHBA), matrix metallopeptidase-11 (MMP11), tumor protein p53 (TP53), and platelet-derived growth factor receptor-beta (PDGFRB) were identified as candidate biomarkers from qPCR analysis, and SPARC and galectin-10 were obtained from the proteomic analysis. The combination of PDGFRB, INHBA, MMP11, and galectin-10 was identified as the optimal combination of survival risk stratification markers. Conclusions: A combination of four proteins in GC tissues may serve as useful survival risk stratification markers in patients with pStage II/III GC following curative resection. Our results may facilitate future multicenter prospective clinical trials.

## 1. Introduction

Gastric cancer (GC) is the fifth most common cancer worldwide and the fourth most common cause of cancer-related deaths [1]. The standard treatment for Stage II/III gastric cancer is postoperative adjuvant chemotherapy after radical resection. This includes S-1 adjuvant chemotherapy, per the Adjuvant Chemotherapy Trial of S-1 for Gastric Cancer (ACTS-GC) [2,3], capecitabine plus oxaliplatin, per the adjuvant study carried out in stomach cancer (CLASSIC trial) [4,5], and S-1 plus docetaxel recommended for pathological (p) Stage III GC based on the results obtained from the JACCRO GC-07 study, where it was shown that the treatment combination improved the survival rate in pStage III GC [6]. 

Although these advances in postoperative adjuvant chemotherapy after radical resection have improved treatment outcomes, with the 5-year survival rate being 80–84.2% for pStage II, 58–67.1% for pStage IIIA, and 50.2–52% for pStage IIIB [3,5]. However, there is a scope for improvement which can be met with personalized therapy based on biomarkers. This includes the selection of personalized postoperative adjuvant chemotherapy based on biomarkers and treatment selection based on risk stratification with respect to recurrence and survival rates.

Several biomarker studies have been carried out where clinical specimens from randomized phase III trials were used to select adjuvant chemotherapy for stage II/III GCs, including the ACTS-GC biomarker study [7,8,9]. In the CLASSIC study, the usefulness of the algorithm in identifying four biomarkers has also been tested using clinical samples [5,10]. In addition, two biomarker studies have also been reported in the Stomach Cancer Adjuvant Multi-Institutional Group Trial (SAMIT) [11,12]. However, though several studies have explored the use of biomarkers for risk stratification based on recurrence and survival in stage II/III GC, only one multiple-cohort study has validated the use of biomarkers in stratifying patients based on survival rate using specimens from the CLASSIC trial [10].

Thus, in this study, we first examined the expression of genes and proteins in clinical specimens obtained from two cohorts of stage II/III GC after curative resection to identify candidate biomarkers. Then we used another large cohort of patient samples to identify the best combination of previously identified candidate biomarkers that would help in stratifying patients based on survival rate.

## 2. Materials and Methods

### 2.1. Search for Candidate Biomarkers for Risk Stratification by Quantitative Real-Time PCR (qPCR) in Patients Who Underwent Curative Resection of GC

#### 2.1.1. Patients and Samples

We included 255 patients with stage II/III GC with no history of neoadjuvant chemotherapy and those who underwent radical GC resection with D2 dissection at the Gastroenterological Center, Yokohama City University Medical Center (145 patients, population 1), and the Department of Gastrointestinal Surgery, Kanagawa Cancer Center (110 patients, population 2), between 2002 and 2009 (Table 1). A fluoropyrimidine-based adjuvant chemotherapy regimen was followed. The Ethics Committees of Yokohama City University (approval number: 18-7A-4) and Kanagawa Cancer Center (approval number: epidemiological study-29) approved the protocol before study initiation, and informed consent was obtained from each patient. GC tissues were obtained from surgical specimens as soon as possible, embedded in optimal cutting temperature (OCT) compound (Sakura Finetechnical Co., Ltd., Tokyo, Japan), and preserved at −80 °C immediately until use. None of the patients had any other malignancy. The histopathological features of the specimens were examined by staining with hematoxylin and eosin, and sections that consisted of more than 80% cancer cells were used to extract total RNA.

#### 2.1.2. RNA Extraction and Complementary DNA (cDNA) Synthesis

Total RNA was isolated from GC tissues using TRIzol (Gibco, Life Technology, Gaithersburg, MD, USA). The cDNA was synthesized from 0.4 µg of total RNA with an iScript cDNA Synthesis Kit (Bio-Rad Laboratories, Hercules, CA, USA) following the manufacturer’s protocol. Subsequently, the cDNA was diluted to 0.2 µg/µL with water and stored at −20 °C until use.

#### 2.1.3. Quantitative Real-Time PCR (qPCR)

qPCR was performed using iQ SYBR Green Supermix (Bio-Rad Laboratories, Hercules, CA, USA), in a total volume of 15 µL, which included 0.2 µg of cDNA, 0.4 µM of each primer, 7.5 µL of iQ SYBR Green Supermix containing dATP, dCTP, dGTP, and dTTP at concentrations of 400 µM each, and 50 units/mL of iTag DNA polymerase. The PCR protocol included 10 min at 95 °C followed by 40 cycles of 10 s at 95 °C (denaturation), annealing for 10 s at a temperature suitable for each gene, and primer extension for 20 s at 72 °C, followed by 10 min at 72 °C. To distinguish specific products from nonspecific products and primer dimers, melting curve analyses were performed. To evaluate the specific mRNA expression in the samples, a standard curve was generated for each run, and three points of the human control cDNA were analyzed (Clontech Laboratories, Inc., Mountain View, CA, USA). The concentration of each sample was calculated using a standard curve.

#### 2.1.4. Gene Selection

In total, 127 genes were analyzed in this study (Figure 1). The following genes were included: genes identified by DNA microarray analysis (details given below), genes and their families targeted by GC-related molecular-targeted agents (including preclinical studies), genes related to the metabolism and activation of GC-related anticancer agents, treatment resistance-related genes, cancer-related genes, cancer stem cell-related genes, 63 genes examined in the ACTS-GC study, and four genes examined in the CLASSIC study [7]. As for gene selection by DNA microarray, the expression of CDH17 and LGALS4 was examined in all cases and histopathologically and genetically classified as intestinal type or diffuse type according to a report by Tan et al. [13]. Furthermore, samples of patients with pathological T3N3M0 stage IIIC disease, and who received standard therapy (curative resection plus adjuvant chemotherapy with S-1) but had recurrence and died in a short period were included in the above intestinal-type and diffuse-type cancer tissues. Comprehensive genetic analysis was performed using the Affymetrix GeneChip^®^ Gene 1.0 ST Array (Thermo Fisher Scientific, Waltham, MA, USA) in both groups of patients to evaluate the expression ratios of approximately 30,000 genes in cancer tissue as compared to the adjacent normal mucosa. Finally, genes that were overexpressed by more than five-fold in intestinal-type and diffuse-type cancer tissues were selected.

#### 2.1.5. Statistical Analysis

In population 1, gene expression cutoff values were determined using the maximum chi-square test, and the relationship between the expression of each gene and the overall survival rate was evaluated. Multivariate Cox proportional hazards analysis was performed to identify the genes that were independent of prognostic factors. Next, these genes were validated in population 2 using the same method as above. Then, genes without inconsistency in hazard ratios (HRs) compared with population 1 and without any bias with respect to the number of patients in subgroups, independent of prognostic factors on multivariate Cox proportional hazards analysis were identified. The reported *p*-values were two-tailed and the statistical significance was set at *p* < 0.05. All statistical analyses were performed using EZR (Saitama Medical Center, Jichi Medical University, Saitama, Japan) [14], a graphical user interface for R (R Foundation for Statistical Computing, Vienna, Austria).

### 2.2. Identification of Candidate Markers for Risk Stratification after Radical Resection of GC by Exploratory Proteome Analysis Using Liquid Chromatography with Tandem Mass Spectrometry (LC-MS/MS)

#### 2.2.1. Patients and Samples

Frozen cancer tissue samples from 24 patients with pT4a NxM0 pStage III GC were obtained from Yokohama City University Hospital. We defined poor prognosis as recurrence within 3 years of curative resection; a good prognosis was defined as survival over 5 years without recurrence following curative resection [15]. The patients were grouped into two groups, the poor outcome group (*n* = 12 patients) and the good outcome group (*n* = 12 patients) (Table 2). Cancer tissues were collected after receiving informed consent from the patients and were preserved at −80 °C until further use.

#### 2.2.2. Sample Preparation for MS-Based Proteomics

Cancer tissues were homogenized in lysis buffer (8 M urea, 4% *w*/*v*) sodium deoxycholate, 10 mM DTT, cOmplete™ Mini, EDTA-free Protease Inhibitor Cocktail (Roche Diagnostics, Mannheim, Germany), and Phosphatase Inhibitor Cocktail 2 or 3 (Sigma, Madison, WI, USA) using a Sample Grinding Kit (GE Healthcare, Piscataway, NJ, USA). The homogenate was sonicated using a UR-21P ultrasonic disruptor (TOMY, Tokyo, Japan) and alkylated with 10 mM iodoacetamide at room temperature for 30 min in the dark. After centrifugation, the sample solution was diluted with 50 mM ammonium bicarbonate to a final concentration of 2 M urea and the proteins were precipitated using acetone. The precipitated proteins were re-dissolved in lysis buffer and diluted 1:4 with 50 mM ammonium bicarbonate. Trypsin (Trypsin Gold, Mass Spec Grade, Promega, Madison, MA, USA) was added to the samples at a protein:enzyme ratio of 20:1 (*w*/*w*), followed by incubation at 37 °C for 16 h. After incubation, sodium deoxycholate was removed using the phase-transfer surfactant (PTS) method from the sample, as previously described [16], and the resulting peptides were desalted and enriched using a self-packed SDB/C18 tip column (Stage tip). Stage tips were prepared by packing Empore SDB XD (3M, Tokyo, Japan) and Empore C18 (3M, Tokyo, Japan) into a 200-µL pipet tip, as previously reported by Rappsilber et al. [17]. The eluted samples were dried completely and stored at 4 °C until LC-MS/MS analysis was performed.

#### 2.2.3. LC-MS/MS Analysis

LC-MS/MS analysis was performed on an LTQ Orbitrap Velos hybrid mass spectrometer (Thermo Fisher Scientific, Yokohama, Japan) with Xcalibur version 2.0.7 and coupled with an UltiMate 3000 LC system (Dionex, LC Packings, Sunnyvale, CA, USA). Relative quantification of label-free proteins was performed using Progenesis QI for proteomics (version 2.0, Nonlinear Dynamics, Durham, NC, USA). To identify the proteins, peak lists were created and searched against human protein sequences in the UniProt Knowledgebase (UniProtKB/Swiss-Prot) database (version May 2013; 538,849 entries) using Mascot (v2.4.1, Matrix Science, Chicago, IL, USA). The search parameters were as follows: trypsin digestion with two missed cleavages permitted; variable modifications: protein N-terminal acetylation, N-terminal carbamylation, oxidation of methionine, carbamidomethylation of cysteine, and phosphorylation of serine, threonine, and tyrosine; peptide charge (2+, 3+, and 4+); peptide mass tolerance for MS data, ±5 ppm; and fragment mass tolerance, ±0.5 Da. We used a false discovery rate of < 1% and a peptide ion score of > 30 as cutoff values to export the results from the MASCOT analysis. The proteins that were differentially regulated between the good and poor outcome groups were extracted using the following parameters in Prognosis QI: maximum fold change > 2 and *p* < 0.001 in one-factor ANOVA. Statistical analysis was performed using GraphPad Prism 5 (version 5.04; GraphPad Software, San Diego, CA, USA).

### 2.3. Tissue Microarray Analysis (TMA)

#### 2.3.1. Patients and Samples

TMA samples were prepared from formalin-fixed paraffin-embedded (FFPE) samples obtained from another cohort of 447 patients with pStage II/III GC who underwent curative surgery and were followed up for >5 years post-surgery (Yokohama City University: *n* = 121, Kanagawa Cancer Center: *n* = 326) (Figure 1). For preparing the TMA, the FFPE samples were punched at the deepest, central, marginal, and adjacent normal mucosa of the tumors. IHC analyses were performed for secreted protein acidic and rich in cysteine (SPARC), human epidermal growth factor receptor 2 (HER2), inhibin subunit beta A (INHBA), matrix metallopeptidase-11 (MMP11), tumor protein P53 (TP53), and platelet-derived growth factor receptor beta (PDGFRB), and galectin-10 (identified as candidate biomarkers by mRNA expression and protein expression analyses).

#### 2.3.2. IHC Analysis Using TMA

The cancer tissue sections were deparaffinized and soaked in 10 mM sodium citrate buffer (pH 6.0) at 121 °C for 15 min for antigen retrieval. After blocking, the sections were incubated overnight with primary antibodies diluted in dilution buffer (PBS of specific pH with 1% BSA, 50% glycerol, and 0.02% sodium azide) at 4 °C. A preliminary examination was performed using positive controls to determine the optimal dilution for each antibody. Anti-SPARC antibody (sc-25574; Santa Cruz Biotechnology, Inc., Dallas, TX, USA) with dilution at 1:50, pH 9; anti-HER2 antibody (#4290; Cell Signaling Technology, Danvers, MA, USA) with dilution at 1:800, pH 9; anti-INHBA antibody (HPA020031; Sigma-Aldrich Co., LLC, Saint Louis, MO, USA) with dilution at 1:200, pH 9; anti-MMP-11 antibody (MS-1035; Thermo Fisher Scientific Inc., Waltham, MA, USA) with dilution at 1:200, pH 9; anti-TP53 antibody (713231; Nichirei Bioscience Inc., Tokyo, Japan) pre-diluted, pH 6; anti-PDGFRB antibody (HPA028499; Sigma-Aldrich Co. LLC, Saint Louis, MO, USA) with dilution at 1:100, pH 6; and anti-galectin-10 antibody (PAC387Hu01; Cloud-Clone Corp., Wuhan, China) with dilution at 1:5000, pH 6 were used. A peroxidase-labeled polymer (EnVision+ rabbit, DAKO, Glostrup, Denmark) and diaminobenzidine were used to detect signals from the antigen-antibody reactions. All sections were counterstained with hematoxylin.

IHC evaluation was performed according to a modified immunoreactivity scoring system (IRS) that categorized immunostaining in the tumor cells based on intensity as absent (score 0), weak (score 1), moderate (score 2), or strong (score 3). Proportion scores were used to classify the proportions of positively immunostained tumor cells into five grades (0, 1, 2, 3, 4, and 5) based on a marker-specific approach. The sum of the intensity and proportion scores ranged from 0 to 8. A score of 0–4 was defined as negative/low protein expression and a score of 5–8 was defined as high protein expression.

#### 2.3.3. Statistical Analysis

When two to seven predictive survival markers were selected among seven candidates and divided into two groups to evaluate the combination for survival risk stratification. The best combination of markers was determined as follows: for each combination, the patients were divided into two groups according to the number of positive markers, and overall survival was analyzed using the log-rank test. Significant combinations in the log-rank test were defined as candidates for predictive survival-marker combinations. Cox proportional hazards analysis was used to perform univariate and multivariate analyses to evaluate the usefulness of the marker combinations. The combination with the lowest *p*-value was determined to be the best combination for predictive survival markers. All statistical analyses were performed using EZR (Saitama Medical Center, Jichi Medical University, Saitama, Japan) [14].

## 3. Results

### 3.1. Identification of Candidate Markers for Risk Stratification

No difference in the clinicopathological features between populations 1 and 2 was observed (Table 1). In population 1, nine genes were selected as prognostic factors using the log-rank test according to the optimal cut-off value (*p* < 0.05). These genes were validated in population 2 using the log-rank test according to the same cut-off value. Genes with a hazard ratio in the same direction (HR > 1 of both populations or both HR < 1) and *p* < 0.05 were adopted as biomarker candidates. SPARC, ERBB2, INHBA, MMP11, TP53, and PDGFRB (six genes) were identified as candidate biomarkers (Table 3). A log-rank test was performed for six genes in 255 patients, and the high expression group showed a significantly worse prognosis relative to the low expression group (Appendix A).

### 3.2. Proteomic Analysis of Candidate Survival Stratification Biomarkers

Proteins (*n* = 1756) common to the poor outcome and good outcome groups (Table 3) were identified using LC-MS/MS. Candidate biomarkers identified based on a maximum fold change > 2 and ANOVA *p* < 0.001 included SPARC and galectin-10 (Table 4). The protein levels of SPARC were higher in the poor prognosis group than in the good prognosis group. Conversely, galectin-10 expression was higher in the good prognosis group than in the poor prognosis group (Appendix A). These trends were maintained in the survival analysis performed using the log-rank test (Appendix A). Additionally, the area under the receiver operating characteristic curve (AUC) indicated that these proteins were useful risk stratification markers (Appendix A).

### 3.3. Identification of the Best Combination of Candidate Survival Stratification Markers

Representative IHC images of tissues positive and negative for SPARC, HER2, INHBA, MMP11, p53, PDGFRB, and galectin-10 expression using IHC with TMA are shown in Figure 2.

We next investigated the usefulness of these seven candidate biomarkers as predictive biomarkers of survival, alone and in combinations. First, we investigated the expression of each protein individually. The high galectin-10 expression group showed a significantly poor survival rate than the low galectin-10 expression group (56.9% vs. 70.1%, respectively; *p* = 0.02; Figure 3). In addition, univariate and multivariate Cox proportional hazards analyses showed that high galectin-10 expression (*p* = 0.03) was an independent prognostic factor (Appendix A). Contrary to the results of mRNA expression and proteome analysis, no statistically significant differences were observed for the other protein markers. On clinicopathological analysis, in the high galectin-10 expression group, the patients were older (*p* < 0.05) than in the low galectin-10 expression group (Appendix A).

We then evaluated combinations of the seven biomarkers. The best combination that could stratify patients included PDGFRB, INHBA, MMP11, and galectin-10 (48.1% (more than two marker-positive among four markers) vs. 70.0% (zero or one marker-positive among four markers); *p* = 0.003) (Figure 4a and Appendix A). ROC analysis showed that AUC was slightly greater than 0.5 for all combinations (Appendix A). Furthermore, analysis in pStage showed that the combination of PDGFRB, INHBA, MMP11, and galectin-10 was more effective in risk stratification, especially in stage III GC (Appendix A). Multivariate Cox proportional hazards analysis showed that when more than two markers were positive among PDGFRB, INHBA, MMP11, and galectin-10 was an independent prognostic factor (*p* = 0.01) (Table 5). Postoperative survival prediction in pStage II/III GC tissue was observed to be superior when expression of this combination of biomarkers was considered together as compared to galectin-10 expression alone. The second-best biomarker combination marker included SPARC, INHBA, MMP11, and galectin-10 (Figure 4b and Appendix A).

Finally, we examined the associations between the expression status of PDGFRB, INHBA, MMP11, and galectin-10 and clinicopathological factors. Significantly higher associations were observed with the histological type (*p* = 0.002) and venous invasion (*p* = 0.023) in more than two marker-positive groups than in the zero or one marker-positive group (Appendix A).

## 4. Discussion

In this study, we aimed to identify useful survival stratification markers for pStage II/III GC after curative surgery. We initially explored the candidate markers by gene expression analysis (qPCR) and proteomic analysis (LC-MS/MS) using tissue samples of stage II/III GC after curative resection, and seven candidate survival stratification markers were identified. Next, we validated the usefulness of these candidate survival stratification markers by IHC analysis of TMAs in another pStage II/III GC cohort. Galectin-10 was identified as an independent significant survival stratification marker amongst the seven marker candidates identified herein. Furthermore, a combination of four markers (PDGFRB, INHBA, MMP11, and galectin-10) was identified as the best survival risk stratification marker combination for pStage II/III GC after curative resection.

Galectin-10 was identified using proteomics analysis and was validated as a significant survival stratification marker using IHC in a different cohort of patients with pStage II/III GC. Contrary to the results of the proteomic analysis, high expression of galectin-10 was observed to be a significantly poor prognostic marker in the IHC analysis. The reason for this discrepancy could be attributed to the difference in the number of patients and patient backgrounds during analyses. Galectin-10 usually resides in the cytoplasm of eosinophils and contributes to extracellular trap cell death, which promotes inflammation and/or efficient pathogen elimination [18]. It aids in the function of CD4^+^CD25^+^Foxp3^+^ regulatory T cells (Tregs), which are immunosuppressive in nature and suppress/downregulate the induction and proliferation of effector T cells [19]. So far, there have been no reports on the relationship between galectin-10 expression in GC tissues and prognosis. Our findings suggest the usefulness of galectin-10 as a survival stratification marker in patients with pStage II/III GC after curative surgery. In addition, galectin-10 may play a key role in GC progression and immunity in tumor-resident eosinophils and Tregs.

INHBA is a member of the transforming growth factor-beta superfamily of proteins that generates activin and inhibin [20,21]. These proteins regulate the hypothalamic-pituitary-gonadal axis [22] and promote cancer progression [23,24]. Moreover, in GC patients, the overexpression of INHBA in tumor tissue is reported to be related to poor survival [25,26,27].

MMP-11 is an enzyme belonging to the MMP family that is involved in the extracellular matrix and tissue remodeling during embryonic development, wound healing, and metamorphosis [28]. Compared to other MMPs, MMP-11 has unique features that may contribute to tumor development and invasion; it is constitutively active and strongly regulates the serine protease inhibitor α1-antitrypsin, insulin-like growth factor binding protein-1 (IGFBP-1), and collagen IV [29,30,31]. The MMP-11 gene and protein expression levels in GC tissues have been reported to be higher than in adjacent normal gastric mucosa [32]. Furthermore, serum MMP-11 levels in tumor tissue and serum have been reported to be useful diagnostic and prognostic biomarkers in advanced GC [33].

SPARC is a matricellular glycoprotein that is generated by many cell types and is related to development, remodeling, cell turnover, and tissue repair [34]. In addition, SPARC has been reported to be involved in the regulation of extracellular matrix turnover and the formation of collagen and MMPs [35]. Although the biological mechanisms of SPARC in GC are not fully understood, it has been reported that high SPARC expression in advanced GC predicts a poor prognosis [36,37]. Furthermore, considering our results that SPARC is a survival stratification marker of pStage II/III GC after curative surgery at both the gene and protein levels, SPARC expression may be related to cancer infiltration and metastasis.

PDGFRB is a cell surface tyrosine kinase receptor for members of the platelet-derived growth factor family and is involved in the regulation of multiple tumor-associated processes including tumor progression, angiogenesis, and regulation of tumor fibroblasts [38]. In addition to our analysis, other studies have suggested that PDGFRB expression level is useful as a prognostic marker in GC [39,40,41].

Although these biomarkers do not interact significantly with each other in biological processes and molecular functions, they are similarly associated with tumor growth, invasion of the tumor, and metastasis. Moreover, our findings further suggest that they can be used as potential biomarkers in survival stratification in GC patients.

This study has some limitations. First, this study only revealed survival risk stratification markers. Hence, cohorts with detailed clinical data need to be investigated for recurrence risk stratification and efficacy markers for adjuvant chemotherapy. Second, RNA and protein were extracted from 5 mm GC tissue sections. Therefore, they may not be representative of RNA/protein expression from the entire tumor, considering the heterogeneity of tumors.

## 5. Conclusions

In conclusion, we showed that a combination of expression of PDGFRB, INHBA, MMP11, and galectin-10 in GC tissues may serve as a useful biomarker for survival stratification in patients with pStage II/III gastric cancer following curative resection. Moreover, this would help in developing personalized adjuvant chemotherapies in the future for pStage II/III gastric cancer patients.

## Figures and Tables

**Figure 1 cancers-14-04427-f001:**
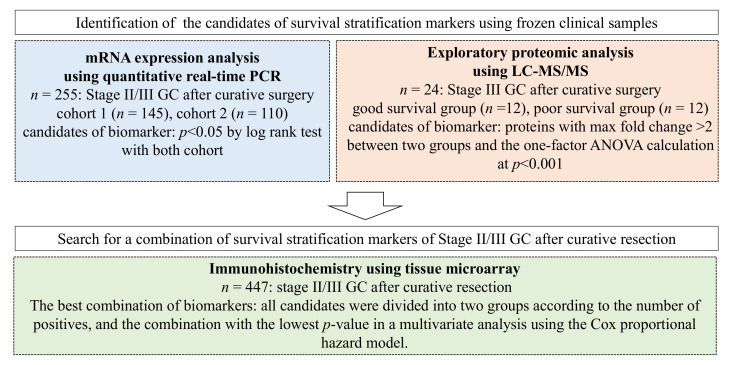
Overview of the strategy for identifying the best combinations of survival risk stratification markers using quantitative polymerase chain reaction (qPCR), liquid chromatography-mass spectrometry (LC-MS)/MS, and immunohistochemistry (IHC) using tissue microarrays (TMAs).

**Figure 2 cancers-14-04427-f002:**
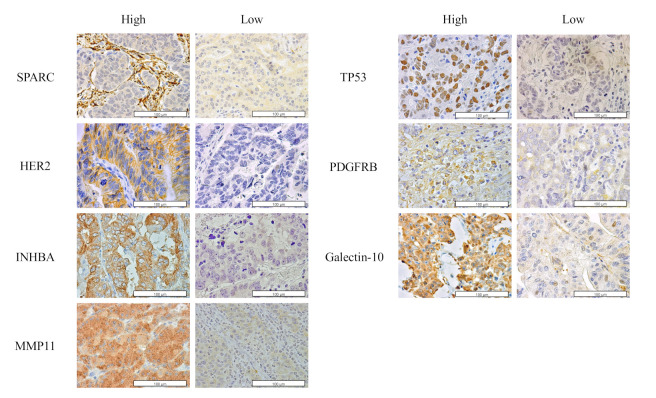
Immunohistochemical status of the candidate survival risk stratification markers secreted protein acidic and rich in cysteine (SPARC), Human Epidermal Growth Factor Receptor 2 (HER2),inhibin subunit beta A (INHBA), matrix metallopeptidase 11 (MMP11), tumor protein P53 (TP53), and platelet-derived growth factor receptor-beta (PDGFRB), and galectin-10 identified by quantitative polymerase chain reaction (qPCR) and proteomic analysis. Representative images of high and low immunostaining of each candidate marker are shown.

**Figure 3 cancers-14-04427-f003:**
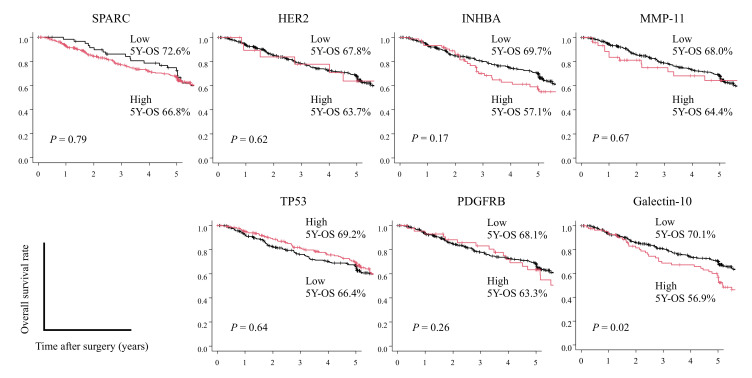
Overall survival rate (OS) of stage II/III gastric cancer (GC) patients according to the expression of each candidate survival risk stratification marker.

**Figure 4 cancers-14-04427-f004:**
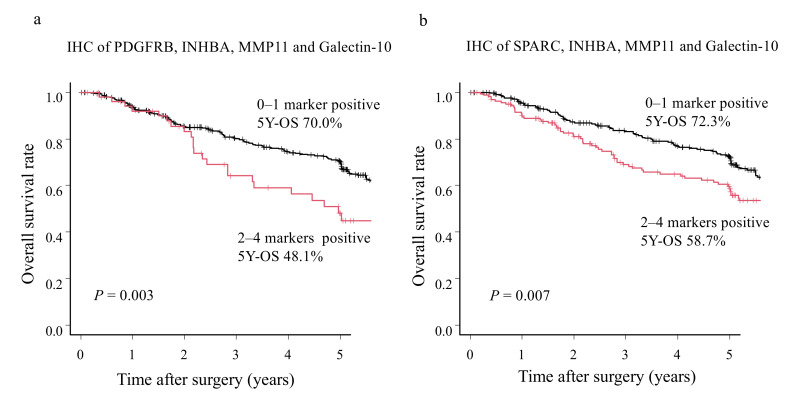
Overall survival rate (OS) of stage II/III gastric cancer (GC) patients according to the combination of survival risk stratification marker expression. A combination of inhibin subunit beta A (INHBA), matrix metallopeptidase-11 (MMP11), platelet-derived growth factor receptor-beta (PDGFRB), and galectin-10 (**a**), and a combination of secreted protein acidic and rich in cysteine (SPARC), INHBA, MMP11, and galectin-10 (**b**) are shown.

**Table 1 cancers-14-04427-t001:** The clinicopathological data of 255 patients with stage II/III GC in qPCR analysis.

Variables/Categories		Population 1 (*n* = 145)	Population 2 (*n* = 110)	*p*-Value
Age	<70 y	77	57	0.8387
	≧70 y	68	53	
Sex	male	95	76	0.5476
	female	50	34	
Histological type	well/moderately/	65	48	0.8496
	poorly	80	62	
Tumor size	<65 mm	76	61	0.6296
	≧65 mm	69	49	
Depth of invasion	pT1	4	4	0.0752
	pT2	19	16	
	pT3	54	25	
	pT4	68	65	
Lymph node metastasis	pN0, pN1	98	76	0.7982
	pN2, pN3	47	34	
venous invasion	(−)	53	20	0.4895
	(+)	92	68	
lymphatic invasion	(−)	46	33	0.3666
	(+)	99	55	
pStage	II	63	31	0.2149
	III	82	57	
Adjuvant chemotherapy	(−)	65	33	0.272
	(+)	80	55	

y = years old; p = pathological; well = well-differentiated adenocarcinoma; moderately = moderately differentiated adenocarcinoma; poorly = poorly differentiated adenocarcinoma.

**Table 2 cancers-14-04427-t002:** Patients’ clinical data from proteomic analysis.

T4a Nx M0, StageIII, Curative Resection and Adjuvant Chemotherapy with S-1 for 1 Year				
Variables/Categories		Recurrence and Death < 3 Years (*n* = 12)	Relapse-Free Survival > 5 Years (*n* = 12)	*p*-Value
Age	(mean SD)	66.25 (8.04)	64.58 (8.11)	0.618
Sex	male	3 (25.0)	4 (33.3)	1.000
	female	9 (75.0)	8 (66.7)	
Histological type	diffuse	8 (66.7)	10 (83.3)	0.640
	intestinal	4 (33.3)	2 (16.7)	
pStage	IIIA	4 (33.3)	2 (16.7)	0.413
	IIIB	1 (8.3)	4 (33.3)	
	IIIC	7 (58.3)	6 (50.0)	
Survival time (days)	(mean SD)	675.42 (249.95)	1931.17 (76.48)	<0.001

SD = standard deviation; pStage = pathological stage.

**Table 3 cancers-14-04427-t003:** The candidate risk stratification markers identified by qPCR analysis.

Gene	Population 1		Population 2	
	Hazard Ratio	*p*-Value	Hazard Ratio	*p*-Value
SPARC	2.651	0.0022	2.876	0.0121
ERBB2	2.632	0.0073	2.245	0.0281
INHBA	5.93	0.0031	2.274	0.0241
MMP11	4.02	<0.0001	2.217	0.0578
TP53	2.386	0.0146	1.704	0.1708
PDGFRB	2.687	0.0261	1.611	0.1954

**Table 4 cancers-14-04427-t004:** Candidate risk stratification markers identified by exploratory proteomic analysis.

Accession	Description	Gene Name	Peptide Number	Anova (p)	Max Fold Change
P09486	SPARC	SPARC	1	0.00059	2.32
Q05315	Galectin-10	CLC	1	0.00038	2.61

**Table 5 cancers-14-04427-t005:** Univariate and multivariate analyses of clinicopathological factors and risk stratification markers expression for overall survival.

Factors		Number of Patients	Univariate		*p*-Value	Multivariate		*p*-Value
			HR	95%CI		HR	95%CI	
Age	≦65	217	1			1		
	>65	230	1.42	1.08–1.88	0.014	1.38	1.04–1.83	0.025
Sex	female	134	1					
	male	313	1.12	0.82–1.51	0.482			
Histological type	well/moderately/	193	1					
	poorly	254	1.07	0.81–1.41	0.643			
Lymphatic invasion	(−)	132	1					
	(+)	315	0.96	0.71–1.29	0.780			
Venous invasion	(−)	128	1					
	(+)	319	1.26	0.93–1.71	0.138			
Stage	II	170	1			1		
	III	277	1.46	1.10–1.94	0.009	1.47	1.11–1.96	0.008
Combination of survival stratification markers *	0–1	394	1			1		
	2–4	53	1.82	1.21–2.72	0.004	1.71	1.14–2.56	0.010

Well = well-differentiated adenocarcinoma; moderately = moderately differentiated adenocarcinoma; poorly = poorly differentiated adenocarcinoma; * = PDGFRB, MMP11, INHBA, and galectin-10.

## Data Availability

The data presented in this study are available on request from the corresponding author.

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
