# Peer review of "Identification of a Biomarker Combination for Survival Stratification in pStage II/III Gastric Cancer after Curative Resection"

_cancers, 2022, doi:10.3390/cancers14184427_

Round 1
Reviewer 1 Report
Dear Authors:
The authors conducted a survival predictive model for pathologic stage II and III gastric cancer and identified that the combination of PDGFRB, INHBA, MMP11, and Galectin-10 was identified as the optimal combination of survival risk-stratification markers. This is a novel model and might interest physicians who treat patients with gastric cancer. However, there are several points needed to be explained more clearly.
1. In table 1, the author collected 255 patients for mRNA expression analysis and compared the clinical and pathologic characteristic between 2 patient cohorts. We suggested the author should also compare the information regarding BMI, neoadjuvant chemotherapy or not, adjuvant chemotherapy regimen and LN dissection level (D1 or D2). These factors were also crucial for gastric cancer patients.
2. Nine gene were identified from population 1 and six genes were confirmed after validation with population 2. The author should present the results of log rank test of nine genes in population 1 and 2. Also, we just wondered why not directly perform log rank test using total 255 patients. In that case, the author can select more meaningful genes correlated with survival.
3. Six genes were identified by qPCR analysis and 2 genes were identified by exploratory proteomic analysis. Table 2 and Table 4 summarized that all these genes were significantly correlated with survival. However, in Figure 3, KM curves of OS were insignificant. How to explain it? Were these genes really significantly relevant to survival?
4. In table 3, the author retrieved 24 specimens for proteomic analysis and defined RFS < 3 years as poor prognosis group and RFS > 5 year as good prognosis group. Are there any references to support the definition of poor and good prognosis? The authors should explain how the cuff-off values were set in method part. Also, the poor prognosis group and good prognosis group should be compared with chi-square test, like table 1.
5. The authors identified 6 genes from qPCR and 2 gene from proteomic analysis. Then, they evaluated combinations of the seven biomarkers and the best combination were 4 genes combination. The author should present the process how they determined the best combination, as well as the ROC curves of the combinations.
6. The authors first identified 6 genes from qPCR and 2 genes from proteomic analysis and then validated all these genes using IHC stain. We just wondered why the method so complex and variable? Can the method in this study consistently use IHC from the beginning to the end?
7. In table S3, after stratifying by 4 gene combination, the patient characteristics were not balanced between high risk and low risk group. These unbalanced characteristics might have significant influences on survival. Thus, it should not conclude that the 4 gene combination is a useful survival stratification markers in patients with pStage II/III GC after curative resection.
Author Response
Point 1: In table 1, the author collected 255 patients for mRNA expression analysis and compared the clinical and pathologic characteristic between 2 patient cohorts. We suggested the author should also compare the information regarding BMI, neoadjuvant chemotherapy or not, adjuvant chemotherapy regimen and LN dissection level (D1 or D2). These factors were also crucial for gastric cancer patients.
Response 1: This study was limited to pStage II/III GC patients who received standard therapy. Thus, all included patients did not undergo neoadjuvant chemotherapy and gastrectomy with D2 dissection. The adjuvant chemotherapy regimen was fluoropyrimidine-based. We agree with your suggestion and have added this information about the included patients. However, because this is historical data, BMI was not recorded. Please accept our apologies.
Changes: ‘We included 255 patients with stage II/III GC with no history of neoadjuvant chemotherapy and those who underwent radical GC resection with D2 dissection at the Gastroenterological Center, Yokohama City University Medical Center (145 patients, population 1), and the Department of Gastrointestinal Surgery, Kanagawa Cancer Center (110 patients, population 2), between 2002 and 2009 (Table 1). A fluoropyrimidine-based adjuvant chemotherapy regimen was followed.’ (lines 81-86)
Point 2: Nine gene were identified from population 1 and six genes were confirmed after validation with population 2. The author should present the results of log rank test of nine genes in population 1 and 2. Also, we just wondered why not directly perform log rank test using total 255 patients. In that case, the author can select more meaningful genes correlated with survival.
Response 2: We examined two patient populations to confirm the robustness of prognostic gene expression. In each population, six genes were candidates for prognostic prediction. We also performed the log-rank test in 255 patients and all genes were deemed potentially useful biomarkers.
Changes: The manuscript now contains the following text and Supplementary Figure 1: ‘A log-rank test was performed for 6 genes in 255 patients, and the high expression group showed a significantly worse prognosis relative to the low expression group (Figure S1).’ (lines 258-260)
Point 3. Six genes were identified by qPCR analysis and 2 genes were identified by exploratory proteomic analysis. Table 2 and Table 4 summarized that all these genes were significantly correlated with survival. However, in Figure 3, KM curves of OS were insignificant. How to explain it? Were these genes really significantly relevant to survival?
Response 2: These results were interpreted as follows: first, the background of the patients differed for mRNA, proteome, and TMA analyses. In the proteome analysis, for example, only 24 cases of long-survivors and poor-survivors were included. Second, divergence in mRNA and protein expressions could explain these results. Furthermore, evaluation by TMA and IHC, which is clinically more relevant, is simple but qualitative.
Changes: The manuscript now contains the following new sentence: ‘Contrary to the results of mRNA expression and proteome analysis, no statistically significant differences were observed for the other protein markers.’ (line 292-294)
Point 4. In table 3, the author retrieved 24 specimens for proteomic analysis and defined RFS < 3 years as poor prognosis group and RFS > 5 year as good prognosis group. Are there any references to support the definition of poor and good prognosis? The authors should explain how the cuff-off values were set in method part. Also, the poor prognosis group and good prognosis group should be compared with chi-square test, like table 1.
Response 3: In general, GC patients experience several recurrence events and death up to 3 years after surgery, and the recurrence-free survival rate over 5 years is very low (20–40%). Accordingly, we defined poor prognosis as recurrence within 3 years of curative resection and good prognosis as survival over 5 years without recurrence. Although there are no reports that support this criterion, we cite a paper on the postoperative prognosis of Stage III GC. In addition, Table 3 was revised to include the results of the chi-square test.
Changes: The manuscript now contains the following new sentence, Table 3, and a citation: ‘We defined poor prognosis as recurrence within 3 years of curative resection; a good prognosis was defined as survival over 5 years without recurrence following curative resection [15].’ (lines 156–158)
‘Table 3. Patients’ clinical data from proteomic analysis.’ (line 271)
‘Fang, W.-L.; Huang, K.-H.; Chen, M.-H.; Liu, C.-A.; Hung, Y.-P.; Chao, Y.; Tai, L.-C.; Lo, S.-S.; Li, A.F.-Y.; Wu, C.-W.; et al. Comparative Study of the 7th and 8th AJCC Editions for Gastric Cancer Patients after Curative Surgery. PLoS One 2017, 12, e0187626, doi:10.1371/journal.pone.0187626.’ (lines 468–470)
Point 5. The authors identified 6 genes from qPCR and 2 gene from proteomic analysis. Then, they evaluated combinations of the seven biomarkers and the best combination were 4 genes combination. The author should present the process how they determined the best combination, as well as the ROC curves of the combinations.
Response 5: We focused on its clinical applicability and usefulness as a prognostic indicator for determining the best biomarker combination. Regarding the determination of combinations, there was a lack of description and analysis, which has now been included. ROC analysis was also performed but the AUC was only slightly greater than 0.5 in both cases.
Changes: The manuscript now contains the following new sentence, Supplementary Figure 3, and Supplementary Table 3: ‘The best combination of markers was determined as follows: for each combination, the patients were divided into two groups according to the number of positive markers, and overall survival was analyzed using the log-rank test.’ (lines 241-243)
‘ROC analysis showed that AUC was slightly greater than 0.5 for all combinations (Table S3).’ (linew 302–303)
‘Figure S3: Overall survival rate (OS) of stage II/III gastric cancer (GC) patients stratified according to the combination of survival risk-stratification marker expression based on the number of positive markers.’ (lines 402–404)
‘Table S3: ROC analysis for the combined expression of survival risk-stratification markers.’ (lines 408-409)
Point 6. The authors first identified 6 genes from qPCR and 2 genes from proteomic analysis and then validated all these genes using IHC stain. We just wondered why the method so complex and variable? Can the method in this study consistently use IHC from the beginning to the end?
Response 6: The most important objective of this study was to determine prognostic factors for GC patients after curative surgery using clinically relevant TMA samples. mRNA expression and proteome analyses were considered useful to examine candidate genes and proteins; however, a consistent IHC analysis would be the most ideal analysis.
Point 7. In table S3, after stratifying by 4 gene combination, the patient characteristics were not balanced between high risk and low risk group. These unbalanced characteristics might have significant influences on survival. Thus, it should not conclude that the 4 gene combination is a useful survival stratification marker in patients with pStage II/III GC after curative resection.
Response 7: We totally agree with your opinion and have revised the Conclusion accordingly.
Changes: The manuscript now contains the following new sentence: ‘A combination of four proteins in GC tissues may serve as useful survival stratification markers in patients with pStage II/III GC following curative resection.’ (lines 37–39)
‘In conclusion, we showed that a combination of expression of PDGFRB, INHBA, MMP11, and Galectin-10 in GC tissues may serve as a useful biomarker for survival stratification in patients with pStage II/III gastric cancer following curative resection.’ (lines 393–395)
Reviewer 2 Report
Thank you for having given to me the opportunity to read this interesting paper
my only concern is the applicability and usefulness of this prognostic classification for improving patients’ survival
Author Response
Point 1: Thank you for having given to me the opportunity to read this interesting paper
my only concern is the applicability and usefulness of this prognostic classification for improving patients’ survival
Response 1: The identification of multiple candidate prognostic markers and combinations of these markers using clinically relevant TMA samples in GC was one of the most important aspects of this study. In the future, post hoc analysis of the candidate prognostic markers using samples from multicenter prospective clinical trials for GC can be expected.
Round 2
Reviewer 1 Report
The authors was revised the most of my concerns appropriately. So, I think this paper could be published now.